# Factors Affecting the Deterioration of the Physical Health Status of Taxi Drivers by Age Group

**DOI:** 10.3390/ijerph19063429

**Published:** 2022-03-14

**Authors:** Jongsun Ok, Kyonghwa Kang, Hyeongsu Kim

**Affiliations:** 1Department of Nursing, College of Nursing, Konkuk University, Chungju 27478, Korea; sokei@kku.ac.kr; 2Department of Nursing, College of Health, Welfare and Nursing, Chungwoon University, Hongseong 32244, Korea; 3Department of Preventative Medicine, School of Medicine, Konkuk University, Seoul 05029, Korea; mubul@kku.ac.kr

**Keywords:** physical health, motor vehicles, behavior that risks health, occupational stress, age groups

## Abstract

With the rapidly aging population, taxi drivers are aging at a fast pace, and competition in the taxi industry is intensifying due to the emergence of various transportation platforms. A descriptive secondary data study was conducted (on a total of 936 subjects) to determine the factors affecting the deterioration of taxi drivers’ physical health status (PHS) according to their age group. The increased incidence of chronic diseases and cognitive decline among taxi drivers aged 55–64 years had the greatest influence on the deterioration of their PHS. Driver obesity was more likely to be related to deterioration of the PHS in the drivers aged 55–64 years (OR: 2.459, <0.001) and 35–54 years (OR: 2.133, <0.001). Among the financial factors, a driver’s income and their number of dependent family members were correlated with the deterioration of the PHS for drivers aged 55 years or over. Therefore, chronic diseases, obesity and cognitive decline were related with deterioration of the physical health status. This suggests that attention should be paid to healthcare policies not only for the elderly aged over 65 years but also those aged 50 to 64 years, i.e., middle-aged people at the beginning of the transition to old age.

## 1. Introduction

Taxis are one of the main means of public transportation and guarantee people’s safety in transit. Traffic accidents caused by taxis can lead to a huge loss of human and material resources at the national level [1,2]. It has been reported that 31.3% of taxi drivers in Korea are over 65 years old [3]. The proportion of elderly drivers increased seven-fold from 3.2% in 2006 to 22% in 2016, with taxis representing the highest proportion of elderly drivers among commercial vehicles [3]. More than 90% of taxi drivers in Korea are middle-aged and elderly, and inevitably, age-related factors often worsen their health status.

In the work environment, employees’ ages have been reported to be related to their wages, job characteristics and the occurrence of accidents [4,5]. The Ministry of Employment and Labor defines workers 55 years of age or older as elderly workers [6]. In the US, the relationship between people’s physical health status and age, in people over 55 years of age, represents a current area of research interest [7]. Yet, does their physical health status have a great impact on older people? 

Age is reported to be a factor that affects not only the health status but also the overall quality of life [8,9]. Since middle age is considered the time when the risk of chronic diseases and cardiovascular disease (CVD) begins to increase [10,11], the risk factors for these must be appropriately managed. The middle-aged and elderly can be characterized as follows. Middle age is generally considered to begin between the ages of 35 and 45 and end by the age of 65 [12,13]. The age group from 65 to 100 years old is defined as old age [14].

First, let us look at their physical characteristics. Through examining the occurrence of CVD at regular checkups for 8491 patients aged 30–74 years, the CVD risk was identified to more than double from the age of 35 [10]. It was also reported that the CVD risk is particularly high in middle-aged people at 40–59 years [11]. Due to sedentary behavior, taxi drivers have a high obesity rate [15], and they are reported as a representative CVD risk group [16,17].

The elderly experience a decrease in physical function and musculoskeletal changes due to aging [18]. Taxi drivers have more knee pain [19] and back pain [20] compared to the general population, and it is reported that there are many health risk factors because they have limited physical activity [21]. Taxi drivers are one of the sedentary jobs that causes fatigue from long hours of work, and these factors exacerbate health problems
[15,16,17,22]. 

In a self-reported study on their perceived health status, middle-aged people responded that their perceived physical health was good [23], while the elderly responded that their physical health was bad [24]. Voluntarily practicing health management behavior was found to have a positive effect on the individual’s perception of their subjective physical health status.

As for socioeconomic characteristics, middle-aged people have reasonable finances due to their careers and positions; however, their expenditures may exceed their income due to parental support, children’s education and marriage [25]. The elderly have a low income level, and their financial preparation for living comfortably in their old age has often been insufficient by the time they finish working [26]. Along with the aging of taxi drivers [27], intensifying competition in the taxi industry, an unstable economic situation and increasing durations and intensities of taxi work [28,29,30] are also leading to various health problems and accidents [31].

Aging is a natural, lifelong process that comes to everyone; however, not everyone goes through the same process [32]. Aging is shaped by the social and physical environment we have lived in throughout our lives and the relationships we have formed, and it takes various paths depending on factors including a person’s family, gender and ethnicity [33]. Although a decline in physical function and cognitive impairment due to aging are unavoidable [34], efforts can be made to minimize the negative impacts through lifestyle changes, social support and health policy application.

Previous studies have sought to determine whether the health status of taxi drivers was associated with the occurrence of traffic accidents or dangerous driving behaviors [27,35]. In these studies, the health status of taxi drivers was mainly based on environmental and occupational factors [22,36,37], and participants’ age was considered a risk factor for accidents [37]. 

In this study, we were more interested in taxi drivers’ self-perceived physical health status, and we sought to investigate the relationships of various factors including physical conditions, such as chronic disease and BMI [22,36], which have been evaluated in previous studies.

For health equity, policies that emphasize universality should be considered first; however, given that the age of most taxi workers in South Korea is middle-aged or older, health-management policies for this workforce should also meaningfully ensure public safety. Therefore, this study aimed to investigate middle-aged and elderly drivers to determine the factors that influence the deterioration of their PHS and, accordingly, present foundational data to support the introduction of appropriate health-management policies.

## 2. Materials and Methods

### 2.1. Study Design and Subjects

A descriptive secondary data study investigated 965 subjects in South Korea (male = 960; female = 5). We used convenience sampling for drivers working for corporate or private taxi firms in Seoul. The total number of corporate and private taxi drivers in Seoul is estimated to be around 50,000. To estimate the number of participants required to achieve statistical significance in the study, G*power 3.1.9.4 was used [38]. 

Statistical power analysis was performed using the following: logistic regression with an alpha of 0.05, two-tailed test of significance, 95% power (1−β) and an odds ratio (OR) of 1.35. Through G*power analysis, the minimum sample size was determined to be 910. Thus, the sample of 965 taxi drivers in this study was sufficient to identify significant differences. The inclusion and exclusion criteria as follows:

Inclusion criteria: Taxi drivers aged over 30 years who had taxi driver’s licenses, actively transported passengers and could read, understand and voluntarily participate in the study.

Exclusion criteria: Taxi drivers diagnosed with dementia or on medication that affected their mental state, as well as retired or hospitalized taxi drivers.

### 2.2. Measurements

Based on previous research on taxi drivers, survey questions were developed by a total of seven people, including five Ph.Ds. majored in preventive medicine, neuroscience, psychiatric nursing, psychology and public administration. To evaluate the validity of the developed survey questions, a preliminary survey was conducted with two taxi drivers, one private and one corporate.

#### Variables

This study used physical (perceived health status, chronic disease and Body Mass Index (BMI)), cognitive and financial characteristics that are consistent with the purpose of this study.

We collected data on physical characteristics, such as determining the drivers’ perceived health status (very good, good, usually, bad and very bad), whether they had felt sick or uncomfortable due to acute or chronic disease, an accident or poisoning in the past two weeks and if they had, then the number of days they had felt that way. 

Furthermore, we asked whether they had a disease that was diagnosed by a doctor and required regular visits to a medical institution and/or was being treated, which affected their heart/circulatory system (hypertension, angina/myocardial infarction or heart failure); endocrine system (diabetes mellitus (insulin), diabetes mellitus (oral hypoglycemic agents) or thyroid disease); respiratory system (chronic obstructive pulmonary disease, asthma or sleep apnea); urinary system (renal failure with dialysis); nervous system (memory impairment, sleep disorder, stroke, peripheral neuropathy or Parkinson’s disease); mental health (depression or panic disorder); eyesight (renal disease, cataracts, glaucoma or color blindness); hearing (impairment/loss); or musculoskeletal system (arthritis or disc). The BMI, a simple index of overweight/obesity was calculated by dividing the weight (kg) by the squared value of the height (m). We classified a BMI of 23–24 kg/m^2^ as overweight and a BMI of 25 kg/m^2^ as obesity [39].

Second, we ascertained their level of cognitive decline compared to one year ago using the Korean dementia screening questionnaire-cognition (KDSQ-C) [40]. The items of this tool can be viewed as more general and universal questions when compared with the existing cognitive function evaluation items to evaluate dementia. This tool consists of a total of 15 items and divided into three subscales of five items each: items 1–5 are the global memory function domain, items 6–10 are other cognitive function and behavior domain, and items 11–15 are the activity of daily living domain (see Appendix A for items). Study subjects were asked to zero is not at all, one point is sometimes, and two points is often. The KDSQ is unaffected by age, gender or education.

Total scores range from 0 to 30, cognitive decline is evaluated by dividing 0–2 points, 3–7 points and 8 points or more. The sensitivity and specificity of this tool for the diagnosis of severe cognitive impairment are 79% and 80%, respectively [40]. The test-retest reliability of this tool is 0.80 [41] and the area under the ROC curve (AUC) for the KDSQ for diagnosing severe cognitive impairment is 0.75 [42].

Third, we investigated financial characteristics, such as the personal income (less than one million won, 100–150 million won, 150–200 million won, 200–250 million own, 250–300 million own and over 300 million won), income satisfaction (very sufficient, sufficient, usually, insufficient and very insufficient) and number of dependent family members (adult and under 18 years old).

### 2.3. Data Collection

After receiving approval from the Institutional Review Board/Ethics Committee of Konkuk University (IRB 7001355-202011-E-127), a survey was conducted of taxi drivers operating private or corporate taxis with the help of the Korea National Joint Conference of Taxi Associations and the Korea National Federation of Taxi Workers’ Unions. After approval was obtained from the representatives of taxi companies and taxi drivers’ unions, a questionnaire was provided to corporate and private taxi drivers through the taxi drivers’ union. 

The subjects understood the purpose of the survey and voluntarily completed the survey in a self-report format. A total of 1000 questionnaires were delivered to taxi drivers, and 965 completed questionnaires were finally collected (the total response rate was 96.5%) through the taxi drivers’ union. Among them, the data for five female taxi driver respondents and 24 taxi drivers who did not provide basic information about their age were excluded. The original data collection was conducted from 22 August to 11 September 2018, and a total of 936 questionnaires were finally analyzed.

### 2.4. Data Analysis

The data were analyzed using the statistical program R (R Foundation for Statistical Computing, Vienna, Austria). Descriptive statistics on physical, financial and cognitive characteristics are displayed as the means and standard deviations or frequencies and percentages. The chi-squared test (x^2^) and ANOVA were used to compare the physical, financial and cognitive characteristics of taxi drivers of different age groups. In addition, multiple regression analysis was used as a factor influencing the deterioration of physical health status. 

Before the analysis, as there were random missing values for each item, and multiple imputation was applied to correct the missing value by using existing data to generate several estimated values for those that are missing. In this way, it is possible to use the complete data without discarding those from respondents where there are missing values, and information on the certainty of the replacement values can be considered. After multiple imputation, multinomial regression analysis was performed (repeating the algorithm 20 times).

## 3. Results

### 3.1. General Characteristics According to Age Group of Taxi Drivers

Table 1 compares the general characteristics of the taxi drivers according to their age group. In terms of their physical characteristics, the number of chronic diseases was highest in the taxi driver group over 75 years of age, with 1.68 ± 1.9 (F = 14.82, *p* < 0.001). Considering particular conditions, hypertension (HTN) was highest in the 65–74-year-old taxi driver group (x^2^ = 38.59, *p* < 0.001) and diabetes mellitus (DM) was the highest in the taxi driver group over 75 years old (x^2^ = 18.90, *p* < 0.001). The number of dependent family members was highest, with 3.40 ± 1.2, in the group of taxi drivers aged 35–54 years and was the lowest, with 2.41 ± 1.3, in the group of taxi drivers over 75 years old (F = 77.31, *p* < 0.001). 

The ratio of those with five or more dependent family members was 15.9% and was the highest among the taxi drivers aged 35–54 (x^2^ = 83.49, *p* < 0.001). A personal monthly income under KRW one million was seen most in the taxi drivers aged over 75 years, as is earned by 11.4% of that age group, while an income rate over KRW three million was highest in the taxi drivers who were 35–54 years of age, as earned by 7.5% of that age group (x^2^ = 47.37, *p* < 0.001). 

In terms of income satisfaction, the group of taxi drivers over 75 years of age had the highest satisfaction, with 11.6% of them reporting that they were satisfied with their work, whereas the highest income dissatisfaction was among taxi drivers aged 35–54, with 73.9% reporting that they were dissatisfied with their work (x^2^ = 30.99, *p* < 0.001). Then, for cognitive characteristics, the group with the highest rate of cognitive decline was the group of taxi drivers aged over 75 years, with such a decline affecting 21.7% and 8.7% of the age group, respectively.

### 3.2. Factors Affecting Deterioration of Physical Health Status According to Age Group of Taxi Driver

Table 2 shows the factors affecting the deterioration of drivers’ physical health status according to their age group. First, we found that having many chronic diseases affected the deterioration of the physical health status, and, in particular, the OR of 55–64-year-old taxi drivers were 2.055 (95% CI, 1.912–2.207). In the case of BMI, being overweight was more likely to a deterioration of physical health status. 

The OR of 55–64 age groups was 2.059 (95% CI, 1.397–3.035), and the OR of the 35–54 year was 1.789 (95% CI, 1.106–2.894), whereas in the groups of taxi drivers aged 65–74 (OR = 0.432, 95% CI, 0.333–0.560) and over 75 years of age (OR = 0.191, 95% CI 0.090–0.401). Obesity was more likely to the deterioration of physical health status in the groups aged 35–54 (OR = 2.133, 95% CI 1.381–3.291) and 55–64 (OR = 2.459, 95% CI 1.687–3.585).

A cognitive decline affected the deterioration of the physical health status of taxi drivers, and in particular, the group that saw the greatest deterioration for this reason was the 55–64-year-old taxi driver group. When the cognitive decline increased by one unit, it was less likely to increase 23.731 times (95% CI, 16.374–34.393).

When it came to financial factors, a personal monthly income of KRW 1–1.99 million more likely to the deterioration of physical health status in the groups of taxi drivers aged 35–54 (95% CI, 8,961,350.0–16,062,770.0) and 65–74 years old (95% CI, 180,899.6–267,780.9). Moreover, in the 35–54-year-old age groups (95% CI, 1,118,204.0–2,140,180.0) and 55–64-year-old age groups ((95% CI, 305,698.0–479,941.2), a personal monthly income of KRW 2–2.99 million more likely to the deterioration of physical health status.

## 4. Discussion

This study suggests that the factors affecting taxi drivers’ deterioration of physical health status differ by age group, and we intend to discuss the factors through this lens of age.

When we studied the BMI as a physical factor, being overweight or obesity was associated with a deterioration of physical health status in the age groups of 35–54 and 55–64, whereas in the groups of taxi drivers over 75 years of age, being overweight or obesity was related to an improvement of physical health status. Previous studies reported that taxi drivers have high rates of excess weight and obesity [43].

However, the effect of being overweight or obesity on the physical health status of taxi drivers, according to their age group, was different. In the case of the elderly who were aged 65 years or older, the BMI decreases with increasing age [44,45], and it has been reported that having an abnormal weight, such as being underweight or obesity, among the elderly, has an important effect on the deterioration of physical function [46,47].

Next, when we examined the patterns of coronary heart disease (CHD) among Japanese taxi drivers, the subjects of this study were male taxi drivers aged 45–60, with an average age of 54.1 years and an average BMI of 25.4 kg/m^2^ [12]. In a study that examined the occurrence of metabolic syndromes among taxi drivers, the proportion of taxi drivers aged 30–59 years was 89%, and among them, it was reported that the proportion of taxi drivers with a BMI of 25 kg/m^2^ or more reached 42% [48]. 

That is to say, in the middle-aged taxi driver group, it appear that a high BMI value increases the risk of CHD or metabolic syndrome. In addition, in this study, we noted that hypertension and diabetes mellitus were the most-represented chronic diseases among all age groups. Referring to the literature, an increase in BMI and chronic diseases, such as HTN and DM, are reported as factors that further increase the risk of CVD [17,43]. Middle age is considered the time when the risk of chronic diseases and cardiovascular disease (CVD) begins to increase [10,11].

Moreover, a report on men with hypertension in the United States found that the incidence rate of uncontrolled hypertension among men aged 20–44 was overwhelmingly higher than that of other age groups [49]. Therefore, we propose that the content and goal of an intervention program to improve drivers’ physical health status requires a differentiated approach based on various characteristics according to the age group of taxi drivers. That is, in the case of the middle-aged, a diet and exercise intervention strategy for active weight loss is needed, and in the case of the elderly, an intervention strategy for maintaining the proper weight and muscle mass, rather than weight loss, is needed.

This study found that cognitive decline was related to a deteriorating physical health status, especially in the 55–64-year-old taxi driver group. Existing studies report that cognitive impairment affects a taxi driver’s driving behavior, along with their physical functioning [22]. However, in this study, cognitive decline was found to directly affect drivers’ physical health status, which is different from the findings of other studies. Cognitive impairment tends to increase with age; however, the previous studies mainly targeted the elderly aged 65 or older [34,50]. 

For middle-aged people aged 55 years and older who are not currently elderly but are in the stage of transition to becoming elderly, cognitive impairments, such as dementia, are also important; however, it is necessary to pay more attention to mild cognitive impairments, which represent a potential risk to driving behavior. This is because a mild cognitive impairment can be difficult to detect, not only in the taxi driver himself but also for his family and colleagues, and the symptoms of a mild cognitive impairment can easily be hidden by the driver for various reasons, such as to continue to bring in an income. 

Therefore, it is necessary to consider their age when evaluating a person’s regular cognitive function is necessary, and we propose that a long-term follow-up study on psychological evaluation according to the degree of cognitive impairment is necessary.

As a result of analyzing detailed items with an KDSQ score of 8 or higher, items with particularly high scores are thought to mean forgetfulness rather than dementia. Forgetfulness, which occurs when there are many things to remember in life, affects attention and concentration and is affected by fatigue [27]. The influence of psychological factors, such as depression and anxiety, on memory cannot be completely excluded. 

In particular, taxi transportation is a service industry that deals with people of various age, occupations, genders and races, and it is difficult to completely rule out the influence of job stress as it is a high-stress job [22]. As this tool is a self-reporting questionnaire, it is necessary to use MMSE [51] or MOCA [3], which is an evaluation of cognitive function by trained researchers in order to diagnose dementia more objectively. We also suggest that expert evaluation, such as by a neurologist is needed to accurately diagnose dementia.

Lastly, as a result of this study, in the case of financial factors, it was found that having fewer dependent family members was associated with the deterioration of physical health status in all taxi drivers aged 55 or over. In the case of the personal monthly income, earning KRW 1–1.99 million in the group of taxi drivers 65–74 years of age was related to a deterioration of physical health status. Moreover, earning KRW 2–2.99 million in the taxi drivers who were 55–64 years of age was related to a deterioration of their physical health status. 

Although financial factors have been reported as important in influencing the deterioration of physical health status in previous studies [22], this study specifically presented the ratio of monthly personal income to the number of dependent family members according to the age group of taxi drivers. According to data from Statistics Korea, as of the fourth quarter of 2019, the number of wage-earning jobholders aged 50–59 years increased by 420,000, while the number of wage-earning jobholders aged 60 years and over increased by 2.5 million [52]. 

In other words, the number of elderly people who are working has increased quantitatively, and the age of participation in work is also rising; however, it is implied that qualitatively, jobs for the elderly may be low-wage and long-hour jobs. It is thought that this increases the financial burden experienced by taxi drivers, leading to a decrease in the number of family members who can financially depend on them and, ultimately, affecting the deterioration of the physical health status of taxi drivers.

Beyond this, it is reported that various factors, such as genetic, physical and social environmental elements influence the health status of older people [53]. Therefore, we proposed that taking a person’s financial vulnerability for granted, without considering other factors, simply because of age, should be regarded as a form of discrimination [33].

Low-income groups report that their access to healthcare services in South Korea is limited due to financial factors [54]. Furthermore, it was reported that socioeconomic factors play a part not only in the progress of physical problems but also in traffic accidents caused by taxi drivers and their driving behaviors [22]. Therefore, in preparing policies to resolve income inequality, it is necessary to consider various factors, such as their income amount and dependent family members according to the age of taxi drivers. 

It has also been suggested that it is necessary to make efforts to revise the taxi fare system that is in place [55] according to the current situation of the taxi industry. To solve the financial problems of the taxi industry caused by the emergence of a new mobility industry, it may also be necessary to introduce a contribution system, as is being implemented in the United States and Australia and use this contribution to support the taxi industry and build transportation infrastructure [56].

This study is meaningful as it investigated the factors affecting the deterioration of drivers’ physical health status according to the age groups of taxi drivers based on the results of a large-scale survey. Nevertheless, since the subjects of this study were taxi drivers over 30 years old working in Seoul, and the proportion of taxi drivers over 75 years old was relatively small, we must be careful about generalizing the research results. Furthermore, for personal income, an income range was used to reduce the burden on respondents when collecting data, and as a result of statistical analysis, the range of odds was relatively wide, and thus caution is needed in interpreting the results.

Yet, based on the findings of this study, we can make the following suggestions. Although the majority of taxi drivers are male and the proportion of female taxi drivers is relatively small, research must determine the factors affecting the physical health status of female taxi drivers. In addition, since the proportions of taxi drivers aged 35–54 and older than 75 years were relatively small in this study, it is necessary to expand the sample to identify the factors affecting the physical health status of taxi drivers of these ages.

Beyond this, the KDSQ, which evaluated cognitive decline in this study, was developed in Korea as a self-report tool for screening for possibility of dementia. However, to assess cognitive impairment, we deem data collection by physicians (or neurologists) and investigators trained to perform cognitive impairment assessments to be necessary.

## 5. Conclusions

In this study, we were interested in taxi drivers’ self-perceived physical health state, and we investigated various factors that affect this, including physical factors, such as chronic disease and BMI, as have been evaluated in previous studies.

Our research found that physical, cognitive and financial factors have various influences on the physical health of taxi drivers in all age groups. In particular, we should pay more attention to various influences on the physical health status of middle-aged people because they are at a pivotal point of health transition that should not be overlooked.

In the taxi industry, where the number of elderly drivers is rapidly increasing due to the aging population, the results of this study suggest a need for continued effort and policy change to improve the physical health of elderly taxi drivers aged over 65. However, the results of this study also suggest that it is necessary to pay attention to healthcare policies for middle-aged people aged 55 and over, as this is the time when the transition to old age begins.

Considering the detailed information that we presented in this article on the factors that affect the deterioration of physical health status according to the age group of taxi drivers, we assert that taxi drivers, who command one of the primary means of public transportation, would benefit from improved policies, such as ones to ensure that their healthcare is equitable, which would support the safety of the public whom they transport every day in South Korea.

## Figures and Tables

**Table 1 ijerph-19-03429-t001:** General characteristics of taxi drivers. n = 936.

Item		35–54 Years	55–64 Years	65–74 Years	≥75 Years	F or x^2^	*p*
		M ± SD or N (%)	M ± SD or N (%)	M ± SD or N (%)	M ± SD or N (%)		
	No. of subjects	135 (14.4)	306 (38.5)	395 (42.2)	46 (4.9)		
Physical	PHS						
	Good	60 (45.1)	168 (46.8)	172 (43.9)	15 (32.6)	9.86	0.131
	Usual	51 (38.4)	156 (43.4)	160 (40.8)	22 (47.8)		
	Bad	22 (16.5)	35 (9.8)	60 (15.3)	9 (19.6)		
	Had PHD	25 (19.1)	42 (12.3)	54 (14.3)	6 (13.6)	3.57	0.312
	Duration of PHD (days)	0.84 ± 2.3	0.58 ± 2.0	0.63 ± 1.9	0.33 ± 1.1	1.21	0.272
	BMI (kg/m^2^)	25.0 ± 2.8	24.9 ± 2.8	24.7 ± 2.7	25.3 ± 3.8	0.35	0.552
	<23	28 (24.56)	82 (23.98)	93 (25.90)	7 (18.92)	5.23	0.515
	23–24	33 (28.95)	91 (26.61)	114 (31.76)	14 (38.84)		
	≥25	53 (46.49)	169 (49.41)	152 (42.34)	16 (23.24)		
	No. of chronic diseases	0.95 ± 1.7	0.90 ± 1.4	1.30 ± 1.6	1.68 ± 1.9	14.82	<0.001
	Types of chronic disease						
	HTN	24 (18.5)	110 (31.2)	178 (45.8)	20 (45.5)	38.59	<0.001
	DM	18 (13.9)	57 (16.2)	100 (25.7)	15 (34.1)	18.90	<0.001
	COPD/asthma	8 (6.2)	14 (4.0)	20 (5.1)	3 (6.8)	1.49	0.684
	Sleep disorder	10 (7.7)	20 (5.7)	22 (5.7)	1 (2.3)	1.88	0.597
	Depression/anxiety	7 (5.4)	9 (2.6)	12 (3.1)	1 (2.3)	2.65	0.449
	Hearing loss	6 (4.6)	11 (3.1)	25 (6.4)	6 (13.6)	10.66	0.014
	Arthritis	17 (13.1)	31 (8.8)	41 (10.5)	7 (15.9)	3.40	0.334
	Slipped disc	13 (10.0)	31 (8.8)	45 (11.6)	9 (20.5)	6.12	0.106
Finances	No. of DFMs	3.40 ± 1.2	2.94 ± 1.2	2.44 ± 0.9	2.41 ± 1.3	77.31	<0.001
	1–2	32 (28.3)	134 (43.0)	234 (67.1)	30 (80.1)	83.49	<0.001
	3–4	63 (55.8)	153 (49.0)	104 (29.8)	4 (10.8)		
	≥5	18 (15.9)	25 (8.0)	11 (3.1)	3 (8.1)		
	Personal income (10,000 won/month)						
	<100	5 (3.8)	11 (3.1)	23 (5.9)	5 (11.4)	47.37	<0.001
	100–199	69 (51.9)	209 (58.7)	279 (71.7)	33 (75.0)		
	200–299	49 (36.8)	121 (34.0)	80 (20.6)	6 (13.6)		
	≥300	10 (7.5)	15 (4.2)	7 (1.8)	0 (0.0)		
	Satisfaction with income						
	Enough	1 (0.7)	11 (3.1)	15 (3.9)	5 (11.6)	30.99	<0.001
	Average	34 (25.4)	110 (30.8)	159 (41.0)	19 (44.2)		
	Not enough	99 (73.9)	236 (66.1)	214 (55.1)	19 (44.2)		
Cognition	Cognitive decline (score)	3.49 ± 4.9	2.98 ± 4.7	3.18 ± 5.1	4.15 ± 5.6	0.13	0.716
	KDSQ 0–2	106 (79.1)	288 (81.4)	317 (80.7)	32 (69.6)	6.45	0.375
	KDSQ 3–7	17 (12.7)	45 (12.7)	43 (10.9)	10 (21.7)		
	KDSQ ≥ 8	11 (8.2)	21 (5.9)	33 (8.4)	4 (8.7)		

No. = number of; PHS = physical health status; PHD = physical health discomfort in last two weeks; BMI = body mass index; HTN = hypertension; DM = diabetes mellitus; COPD = chronic obstructive pulmonary disease; DFM = dependent family members, either adults or underage; and KDSQ (Korean dementia screening questionnaire).

**Table 2 ijerph-19-03429-t002:** Factors affecting deterioration of physical health status according to age group of taxi drivers.

Items	Odds	CI	*p*	Odds	CI	*p*	Odds	CI	*p*	Odds	CI	*p*
	35–54 Years			55–64 Years			65–74 Years			≥75 Years		
No. of chronic diseases	1.551	1.412–1.705	<0.001	2.055	1.912–2.207	<0.001	1.610	1.525–1.700	<0.001	1.550	1.292–1.859	<0.001
BMI (Reference: <23)												
Overweight (23–24)	1.789	1.106–2.894	0.018	2.059	1.397–3.035	<0.001	0.432	0.333–0.560	<0.001	0.191	0.090–0.401	<0.001
Obesity (≥25)	2.133	1.381–3.291	<0.001	2.459	1.687–3.585	<0.001	1.095	0.891–1.346	0.387	0.441	0.227–0.855	0.015
Duration of PHD	1.544	1.395–1.710	<0.001	1.046	0.985–1.112	0.141	0.946	0.901–0.993	0.025	0.958	0.559–1.542	0.774
No. of DFMs	1.080	0.929–1.255	0.319	0.869	0.764–0.988	0.033	0.803	0.729–0.885	<0.001	0.901	0.729–1.115	0.337
Personal income (10,000/month) (Reference: <100)												
100–199	11,997,670.0	8,961,350.0–16,062,770.0	<0.001	216,386.7	169,116.3–276,870.0	<0.001	220,094.2	180,899.6–267,780.9	<0.001	98.642	58.694–165.779	<0.001
200–299	1,546,984.0	1,118,204.0–2,140,180.0	<0.001	383,036.6	305,698.0–479,941.2	<0.001	222,143.7	191,513.5–257,672.8	<0.001	36.625	26.509–50.601	<0.001
Income dissatisfaction (Reference: Enough)												
Not enough	0.814	0.536–1.236	0.334	0.479	0.348–0.659	<0.001	0.865	0.712–1.050	0.143	0.920	0.498–1.697	0.789
Cognitive decline (Reference: KDSQ 0–2 score)												
KDSQ 3–7 score	1.530	0.942–2.486	0.086	10.836	8.088–14.518	<0.001	5.048	4.086–6.237	<0.001	2.962	1.586–5.533	<0.001
KDSQ ≥8 score	13.826	7.967–23.993	<0.001	23.731	16.374–34.393	<0.001	3.343	2.544–4.394	<0.001	4.845	1.846–12.714	0.001

BMI = body mass index; PHD = physical health discomfort within two weeks; No. = number of; DFM = dependent family members; KDSQ (Korean dementia screening questionnaire).

## Data Availability

Not applicable.

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
