# Peer review of "Factors Affecting the Deterioration of the Physical Health Status of Taxi Drivers by Age Group"

_ijerph, 2022, doi:10.3390/ijerph19063429_

Round 1

Reviewer 1 Report

Effects of the deterioration of physical health status of taxi drivers by age group

  • Manuscript ID: ijerph-1578162
  • Journal: International Journal of Environmental Research and Public Health

The current study by Ok JS. et al reports the factors influencing the physical health status (PHS) of taxi drivers according to their age distribution. Although, the current study is based on a large scale relatively, still its results cannot be implemented as a general inference. As authors agree that the number of elderly age taxi drivers is increasing with time, it will be important to include this age group members as much as possible to conclude a more specific and precise outcome. Main strength of this study is to correlate the PHS of drivers according to their age group. It is also very critical to pay attention to improve the health policies for the middle age group drivers, as they are transitioning towards elderly age. This kind of study can help to re-evaluate the public health policies for the taxi drivers and other professions too.

  • Abstract and Introduction parts are well-structured.
  • There is sufficient connection between literature survey and study design.
  • Methodology described looks sufficient but can be improved in future large scale research.
  • Discussion and conclusions are described in a good way.
  • Overall, manuscript is well-written.

Author Response

Dear reviewer

There is nothing to respond because there are no special comments.

Sincerely

Kyonghwa Kang

Reviewer 2 Report

The manuscript, “Effects of the deterioration of physical health status of taxi 2 drivers by age group” aimed to explore how age was associated with physical health among Taxi drivers in South Korea.  To improve the manuscript, I would encourage the authors to consider the following.

  • I would start the manuscript by discussing why driving a taxi matter. I would consider beginning the manuscript with paragraph two of page two, which begins with “Taxis are of the main means of public transportation. . .. “
  • I would consider changing some of the language used, which implies causation. I would consider using more words such as “associations” or “correlations,” where appropriate.
  • The study was A-theoretical, which makes hard to determine how this deteriorating health was viewed. It also makes it harder to determine what variables to include in the study. A framework would enhance how human behavior among taxi drivers should be viewed.
  • Was this a cross-sectional survey research design? If so, that should be stated in the manuscript, instead of the purpose of the study.
  • Was sampling method was used?
  • The authors should provide validity and reliability information on the measured used in this study.
  • Were there any ethnic groups who participated in this study?
  • The recruitment methods should be detailed a bit more clearly.
  • What was the age range for drivers under the age of 55?
  • How much missing data was present?
  • I wonder if the study has enough statistical power to detect statistical significance among respondents who 75 and over? This can lead to a Type II error.
  • The authors should change the word on line 172 of page 8, “This study found that” to “This study revealed that” or “This study suggests that.”
  • The manuscript should include future research questions.

Author Response

Thank you very much for your valuable time and for your excellent comments for our manuscript.

Sincerely,

Kyonghwa Kang

Reviewer 3 Report

Dear Authors,

Your article entitled “Effects of the deterioration of physical health status of taxi drivers by age group” is interesting, but unfortunately it cannot be considered now for possible publication in IJERPH, as various major revisions are needed, as listed here below for the different sections of the manuscript.

Best regards,

The Reviewer

Abstract

  • The length of the abstract should be reduced according to the journal’s rules.
  • Please carefully revise English language of the abstract: some sentences are not clear/grammar not correct.

1.Introduction

  • Lines 27-32: I am not sure these premises are correct. It seems to me that the age range 30-64 years is too wide. Why people in their thirties should be considered at risk for chronic diseases? I don’t think so. The references you cite to support these statement are not authoritative, referring to minor articles published in journals with a moderate/low IF. Please, see official data, reports and publications of authoritative institutions as e.g. WHO, CDC, ECDC, international scientific societies dealing with the problem of aging and chronic diseases. Please see e.g.: https://www.who.int/teams/social-determinants-of-health/demographic-change-and-healthy-ageing/combatting-ageism/global-report-on-ageism
  • What are “negative” and “positive” subjective health status?
  • Line 52: yes, it is somewhat unavoidable, but preventive actions can be taken to reduce the negative effects of aging on the decline of the health status, in particular acting on lifestyles, social support and health determinants.
  • Lines 55-56: I don’t believe there are no existing studies categorizing the taxi drivers by age group. Please check the literature again. See e.g.: 10.3233/WOR-131696 and various others
  • Lines 59-61: again, please check scientific literature. There are studies of interes. E.g. 10.3233/WOR-131696 , 10.3233/WOR-131696 and others

2.Materials and methods

  • 1. Study Design: line 71. Do you mean observational study?
  • 2. Study Subjects: more details on the target population, inclusion and exclusion criteria, number of potential participants are needed.
  • 3. Measurements: this is a too generic description. Please provide more details on the questionnaire and the items, referring also to previous research if the questionnaire has been already used. Consider also to add the full questionnaire as a supplementary material.
  1. Results

- I have a fundamental doubt here: I am not a neurologist or a psychiatry specialist, but I am a physician and I know what is dementia. I don’t think that a subject affected by dementia can drive. And in my country, as soon as you have symptoms of dementia, the system revokes your drive license, especially if you are a professional driver. The percentages of drivers with dementia are too high, in my opinion, to be coherent with an appropriate definition of dementia, responding to adequate criteria, considering that the population studied is an active population of taxi drivers, and not a population including retired or hospitalized subjects. Please explain in detail how you got these results and the criteria applied to recognize dementia.

  1. Discussion

- Line 185: please consider that the increased CVD risk can be related also to high job stress during professional driving as reported in this research on bus drivers: http://dx.doi.org/10.3233/WOR-172581

  1. Conclusions

- The role of “age categorization” should not be emphasized as a strength of the study: it is a minimum requirement of a good study, the majority of the studies usually divide subjects per age groups and moreover your categorization is not so detailed (i.e. <55 ys?)

  1. References

- Please carefully check scientific literature to find other relevant studies on professional drivers (the suggested ones and various others) as well as international authoritative documents on the problem of aging & health and on the definition and diagnostic criteria for dementia.

Author Response

Thank you very much for your valuable time and for your excellent comments for our manuscript. Please see the attachment.

Sincerely,

Kyonghwa Kang

Round 2

Reviewer 3 Report

Dear Authors,

unfortunately your answers to my comments were too much generic and did not explain many of the concerns I raised.

Again, the main issue, among several others, is related to the definition of dementia. According to your results, more than 8% of young taxi drivers between 34-55 years old have a "severe cognitive impairment" (SCI). SCI is "a deterioration or loss in intellectual capacity that places a person in jeopardy of harming him or herself or others and, therefore, the person requires substantial supervision by another person. The condition is measured by clinical evidence and standardized tests which reliably measure impairment in short or long term memory, orientation to people, places or time, and deductive or abstract reasoning (https://en.wikipedia.org/wiki/Severe_cognitive_impairment#:~:text=Under%20the%20United%20States'%20Federal,supervision%20by%20another%20person%3B%20and).

For me it is unbelievable that a percentage between 6 and 9% of taxi drivers in your country has this condition, which is, to me, completely incompatible with work activity of taxi drivers. Perhaps there is a "cultural" issue related to the questionnaire used and the definition of SCI in your country. Moreover, there can be limitations related to the study design possibly overestimating the prevalence of SCI in the sample of taxi drivers. 

In any case, also according to these considerations my suggestion is to re-submit your article to a journal with a less international public, perhaps addressing interested readers from your Country and similar Countries, otherwise the results you are going to present could be misinterpreted, leading to wrong conclusions and generalizations.

My apologies for the decision of rejecting your paper,

with kindest regards,

the Reviewer

Author Response

Dear Reviewers

You gave us a valuable opportunity to think about how to present the result of research.

Thanks to your comments, we think our research paper has improved significantly.

We tried to explain the KDSQ tool in more detail among the study methods. Also, the items of the KDSQ tool are presented in appendix A.

We did not use terms such as dementia or cognitive impairment, as we thought that the KDSQ tool was to evaluate universal contents. We also added to the discussion that a score of 8 or higher on the KDSQ tool may be associated with forgetfulness due to fatigue, emotional problems, and stress in younger age groups in taxi drivers.

Please address all correspondence concerning this manuscript to me at [email protected]

Best regards,

Kyonghwa Kang